# GAUSSIAN ENTROPY FLOW WORLD MODEL FOR STREAMING 3D OCCUPANCY PREDICTION

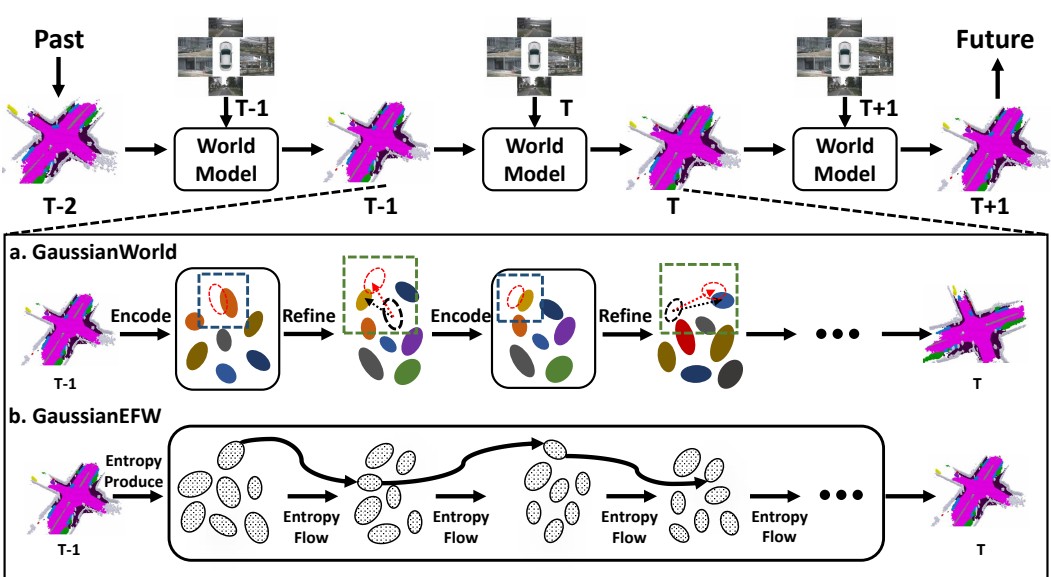

Figure 1: (a) GaussianWorld models continuous scene evolution, but discrete refinements over multiple encoding–decoding steps cause cumulative errors. Blue, green, and red dashed lines show encoding errors, refinement errors, and ground-truth position. (b) GaussEFW represents scene evolution as a single continuous flow of Gaussian Entropy in latent space. Predicting this flow from current RGB observations allows accurate modeling of continuous scene evolution.

## ABSTRACT

In 3D occupancy prediction, temporal information is crucial. Traditional methods fuse multi-frame features through a pipeline of perception, alignment, and fusion, but they overlook the coherence of static elements and the motion patterns of dynamic elements in 3D scenes. Existing methods reformulate 3D prediction as 4D prediction based on current sensor inputs by modeling the continuous evolution of the scene. However, the discrete refinements of the physical properties of dynamic elements in multiple encoding-decoding processes lead to cumulative errors and poor adaptation to dynamic motion. Inspired by non-equilibrium thermodynamics, we propose an Evolutionary Entropy Flow framework that uses Evolutionary Entropy as a carrier for continuous scene evolution, modeling the motion of dynamic elements as the flow of Evolutionary Entropy. We further introduce the Gaussian Entropy Flow World model (GaussEFW), which represents Evolutionary Entropy Flow as a single, continuous Gaussian Entropy Flow in latent space, in contrast to the discrete refinements from multiple encoding-decoding processes. By predicting Gaussian Entropy Flow based on current RGB observations, we can accurately predict the motion of dynamic elements and learn continuous scene evolution. Extensive experiments on the nuScenes dataset validate the effectiveness of GaussEFW, demonstrating superior performance in dynamic element prediction and high overall performance.

# 1 INTRODUCTION

3D semantic occupancy prediction (Wei et al., 2023) is a fundamental component of autonomous driving, as it jointly infers occupancy and semantics from visual observations to construct fine-grained representations of the environment (Huang et al., 2024b; Li et al., 2023; Xia et al., 2023). A central challenge lies in modeling scene dynamics (Ma et al., 2024; Liu et al., 2023; Wang et al., 2022; Jin et al., 2024; Li et al., 2024), which requires exploiting temporal continuity across frames. While successive frames are inherently correlated through ego-motion and element movements, most existing perception and fusion approaches fail to leverage these correlations effectively (Wang et al., 2023b). Naive feature fusion not only neglects the stability of static structures and the motion regularities of dynamic entities but also introduces considerable computational overhead. To overcome these limitations, recent works propose representing the scene with 3D gaussian primitives (Kerbl et al., 2023; Huang et al., 2024c;a) and reformulating occupancy prediction as a 4D problem that integrates static alignment, dynamic motion, and novel observations (Zuo et al., 2024). By employing a single refinement module to jointly update historical and newly observed Gaussians, such approaches capture temporal coherence more effectively, thereby improving scene understanding while reducing the complexity and cost of temporal modeling.

The Gaussian World Model (Zuo et al., 2024) demonstrates strong performance in static environments but exhibits notable limitations in dynamic scenarios. Specifically, the model refines the physical attributes of dynamic elements through repeated encoding–decoding cycles across transformer blocks (Huang et al., 2024c;a). This design introduces several challenges. As shown in the figure 1, dynamic elements, unlike static structures, possess intrinsic mobility, and iterative encoding–decoding operations introduce cumulative positional deviations that propagate into subsequent state predictions, compromising overall accuracy. Moreover, accurately representing these elements requires higher-resolution geometric and semantic features, yet repeated encoding–decoding progressively erodes such details, diminishing representation fidelity. The resulting compounded errors further distort the latent feature embeddings of dynamic elements, hindering precise attribute refinement and amplifying error propagation throughout the predictive process.Collectively, these issues undermine the model's adaptability to dynamic environments and degrade predictive accuracy, thereby constraining its applicability to complex real-world scenarios.

Inspired by nonequilibrium thermodynamics, we introduce an evolutionary entropy flow framework that models scene dynamics through evolutionary entropy as the carrier of continuous evolution. In this formulation, the evolution of dynamic foreground elements is represented as a continuous entropy flowing in latent entropy space. The framework consists of three key processes: Entropy Producing, which encodes dynamic elements into the entropy space and injects evolutionary entropy; Entropy Exchanging, which incorporates external observational information to guide the flow direction; and Entropy Flowing, which evolves the entropy toward a stable state under observational guides. By predicting flow of evolutionary entropy, the model enables accurate forecasting of foreground element attributes and achieves spatiotemporal evolution of continuous scenes, particularly capturing the continuous motion of dynamic elements.

We propose the Gaussian Entropy Flow World Model (GaussEFW), which is built upon the evolutionary entropy flow framework and leverages the parameter adjustability, non-structurality, and scalability of gaussian representations (Kerbl et al., 2023; Huang et al., 2024c) to effectively model the continuous motion of dynamic foreground elements and scene evolution (Zuo et al., 2024). Unlike Gaussian World (Zuo et al., 2024), which represents scene evolution as discrete Gaussian movements in physical space via transformer blocks, GaussEFW models the continuous evolution of scene as a single flow of Gaussian Entropy in latent space, naturally capturing complex dynamic interactions and continuous motion patterns.

To achieve accurate prediction, GaussEFW employs an in-model denoising network (Zhou et al., 2025) to iteratively update gaussian queries (Huang et al., 2024c), combined with variable attention mechanisms to perceive observational information, enabling fine-grained modeling of dynamic element attributes. This design not only reduces error accumulation and preserves geometric and semantic details, but also unifies multi-frame information in latent space, allowing precise prediction of foreground element properties and spatiotemporal scene evolution. Through these mechanisms, GaussEFW provides an efficient and robust approach for continuous 3D scene prediction, effectively capturing dynamic element motion and scene evolution in complex environments.

Our main contributions are as follows:

- We propose an evolutionary entropy flow framework to model the continuous evolution of dynamic foreground elements via three processes—Entropy Producing, Entropy Exchanging, and Entropy Flowing—capturing geometric and semantic changes over time.

- We introduce the Gaussian Entropy Flow World Model (GaussEFW), which represents scene dynamics as a single continuous flow of Gaussian Entropy in latent space, reducing error accumulation and preserving fine-grained scene details.

- Extensive experiments on the nuScenes dataset validate the effectiveness of GaussEFW, demonstrating superior performance in dynamic element prediction and achieving high overall performance.

## 2 RELATED WORK

### 2.1 3D TEMPORAL OCCUPANCY PREDICTION

Utilizing temporal information is crucial for 3D perception (Li et al., 2022; Huang et al., 2021; Li et al., 2025; Philion & Fidler, 2020). A common approach is to fuse multi-frame scene representations to enhance perception tasks. These methods (Wang et al., 2023b; Liu et al., 2024; Huang & Huang, 2022) align multi-frame representations to the current time and aggregate temporal information, enhancing temporal prediction accuracy. However, these designs fail to consider the continuity and simplicity of driving scenes, limiting the performance of temporal modeling. Some attempts (Wang et al., 2023a) propose a novel element-oriented temporal modeling mechanism for streaming 3D prediction. Since it uses element queries as scene representations, it can only implicitly model the motion of dynamic elements and is not suitable for dense occupancy prediction (Pan et al., 2024; Tian et al., 2024; Zhang et al., 2023a).

### 2.2 WORLD MODELS IN AUTONOMOUS DRIVING

World models (Ha & Schmidhuber, 2018; Assran et al., 2023; Hafner et al., 2020) are typically defined as models that predict the future based on historical observations and actions. Currently, their applications in autonomous driving primarily include driving scene generation, planning (Gao et al., 2023; Hu et al., 2023; Wang et al., 2024b), and representation learning. World models based on advanced generative models can generate diverse driving sequences (Zheng et al., 2023; Wei et al., 2024). By jointly modeling scene evolution and ego-motion, world models can learn effective driving strategies to support planning tasks (Zheng et al., 2023; Wang et al., 2024a). Additionally, world models have been used for 4D pre-training to acquire general scene representations. Recently, Gaussian World Models (Zuo et al., 2024; Zheng et al., 2024) have leveraged scene evolution learning combined with current RGB observations to enhance 4D occupancy prediction, demonstrating the applicability and benefits of world models for perception tasks.

### 2.3 IN-MODEL DENOISING LEARNING

DDPM (Ho et al., 2020) improves image quality and diversity through multiple decoder iterations, but suffers from slow inference and high resource consumption. DeTrack (Zhou et al., 2025) proposed an in-model implicit variable denoising paradigm, using a single forward pass of Denoising ViT for efficient element tracking. These works collectively demonstrate the advantages of denoising networks in enhancing model robustness and performance. Based on its efficiency, we adopt this paradigm to model the evolution of continuous scenes.

## 3 METHOD

### 3.1 CONTINUOUS SCENE EVOLUTION WORLD MODEL

**Temporal Perception Model** In autonomous driving, the perception model $A$ takes sensor inputs from the current frame $T$ and the past $t$ frames, $\mathbf{o}^T, \mathbf{o}^{T-1}, ..., \mathbf{o}^{T-t}$, along with the corresponding

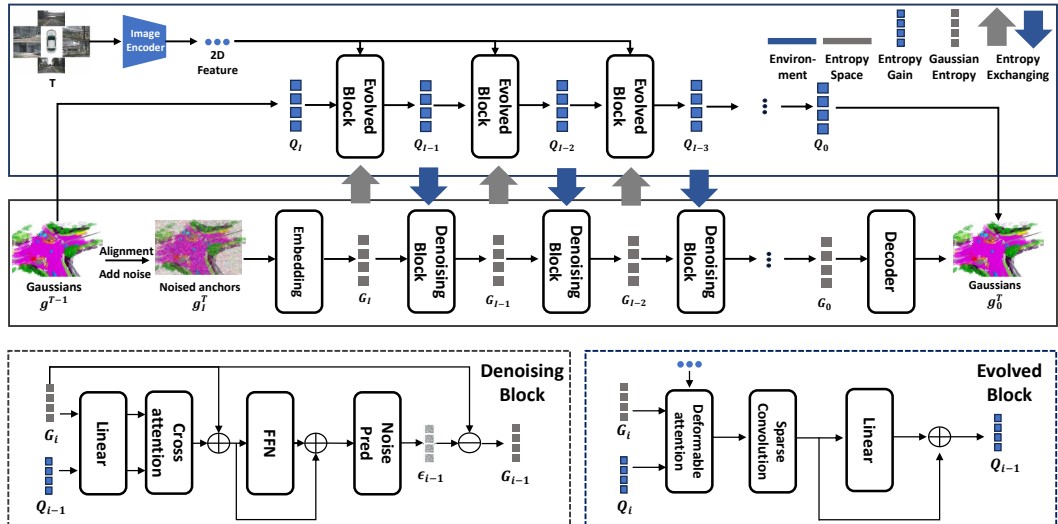

Figure 2: The GaussEFW architecture. Gaussian representations from the previous timestep are aligned and noise is added to the anchors. The Entropy Producing process encodes them into latent space to form Gaussian Entropy, which interacts with the environment during Entropy Exchanging to obtain refined Entropy Gains. As these gains continuously self-refine, Gaussian Entropy gradually flows toward a stable state under environmental guidance.

ego-vehicle poses $\mathbf{p}^T, ..., \mathbf{p}^{T-t}$, to generate the perception output $\mathbf{y}^T$:

$$\mathbf{y}^T = A(\mathbf{o}^T, ..., \mathbf{o}^{T-t}, \mathbf{p}^T, ..., \mathbf{p}^{T-t}) \tag{1}$$

Conventional methods (Wang et al., 2023b; Huang & Huang, 2022) use a three-stage pipeline: the perception module extracts frame-level features $\mathbf{f}$, the transformation module aligns historical features using ego-motion to produce aligned features $\mathbf{a}^t$, and the fusion module integrates the aligned features into a unified scene representation.

$$\mathbf{f}^t = P_{er}(\mathbf{o}^t), \quad \mathbf{a}^t = T_{rans}(\mathbf{f}^t, \mathbf{p}^t), \quad \mathbf{y}^T = F_{use}(\mathbf{a}^T, ..., \mathbf{a}^{T-t}) \tag{2}$$

Although effective, this design overlooks the temporal continuity of driving scenes. Adjacent frames are inherently correlated, which simple multi-frame fusion cannot fully exploit. To address this, we introduce a world model (Zuo et al., 2024; Zheng et al., 2024) $\mathbf{w}$ that refines the current scene representation $\mathbf{r}^T$ from the previous state $\mathbf{r}^{T-1}$ and current observation $\mathbf{o}^T$, and outputs the perception $\mathbf{y}^T$ via the perception head $\mathbf{h}$.

$$\mathbf{r}^T = \mathbf{w}(\mathbf{r}^{T-1}, \mathbf{o}^T), \quad \mathbf{y}^T = \mathbf{A}(\mathbf{r}^{T-1}, \mathbf{o}^T) = \mathbf{h}(\mathbf{w}(\mathbf{r}^{T-1}, \mathbf{o}^T)) \tag{3}$$

This formulation enables the model to learn the joint distribution of scene states conditioned on both temporal evolution and current observations.

## 3.2 EVOLUTIONARY ENTROPY FLOW

**Evolutionary Entropy.** While this method performs well in static scenes, it encounters challenges in dynamic environments. Repeated encoding–decoding between consecutive states leads to the gradual loss of fine geometric and semantic details of dynamic elements and results in error accumulation. Inspired by nonequilibrium thermodynamics, we propose an evolutionary entropy flow framework that models the motion of dynamic elements as entropy evolution in high-dimensional space and represents continuous scene changes as entropy flows gradually converging to stability under environmental guidance.

**Entropy Producing.** To enable the continuous evolution of dynamic scene elements, we introduce the notion of evolutionary entropy, which serves as a carrier for modeling the dynamics of elements in high-dimensional entropy space. Formally, the evolutionary entropy of a probability distribution $p(x)$ , with $x$ as the system state variable,is defined as:

$$S = - \int p(x) \log p(x) \, dx. \tag{4}$$

To model the entropy production of evolutionary entropy, we apply controlled stochastic perturbations to the aligned previous state, mapping it into a high-dimensional latent space (entropy space) to capture the potential variations of dynamic elements. Given the aligned state $r^{T-1}$, the initial evolutionary entropy $S_I$ is computed as:

$$r_I^T = N_{oise}(r^{T-1}), \quad S_I = \mathcal{E}_{\mathcal{P}}(r_I^T) \tag{5}$$

Where $N_{oise}$ maps aligned $r^{T-1}$ to the perturbed state $r_I$, and $\mathcal{E}_{\mathcal{P}}$ encodes $r_I$ into the initial evolutionary entropy $S_I$ in the entropy space, as shown in Equation 4.

**Entropy Exchanging.** Once initial evolutionary entropy is generated via Entropy Producing, the entropy space interacts with environmental observations to guide its evolution. For a given evolutionary state $S_i$, its interaction with the observation $o^T$ updates the associated Entropy Gain $c_i$ through the mapping $c_{i-1} = \mathcal{E}_{\mathcal{E}}(S_i, c_i, o^T)$, which then directs the subsequent evolution of the entropy. As the environment changes, the evolutionary entropy continuously adapts within the entropy space, capturing the dynamic influence of external information. Details are provided in the appendix.

**Entropy Flowing.** Guided by the Entropy Gains generated during Entropy Exchanging, the evolutionary entropy gradually evolves toward a stable state. We first predict the target state $r_0$ using a probabilistic model. Following the Markov principle, the process can be decomposed into successive steps, and the probability of the target state $r_0$ is

$$p(r_0) = p(r_I) \prod_{i=1}^{I} p(r_{i-1} \mid r_i, c_i), \tag{6}$$

which represents the probability of the target state as the product of the initial state's probability and a series of conditional probabilities under the Entropy Gains $c_i$.

According to the definition of evolutionary entropy in Eq. equation 6, the stable-state evolutionary entropy can be compactly written as

$$S_0 = S_I - \sum_{i=1}^{I} S_i', \quad S_i' = \int p(r_{i-1} \mid r_i, c_i) \log p(r_{i-1} \mid r_i, c_i) \, dr_{i-1}. \tag{7}$$

By predicting the evolutionary entropy flow $S_i'$, the evolutionary entropy decreases monotonically, ultimately converging to the stable state $r_0$ of scene evolution. The detailed derivation is provided in the Appendix.

### 3.3 GAUSSIAN ENTROPY FLOW WORLD MODEL

**Gaussian Representation.** We represent the scene with a set of sparse 3D semantic Gaussians (Kerbl et al., 2023; Chambon et al., 2025; Huang et al., 2024c), each encoding position, scale, rotation, semantic probability, and a temporal feature capturing historical information. We predict current gaussian representations from the previous state $g^{T-1}$ and current observation $o^T$, aligns previous-frame gaussian representations to account for ego-motion via an affine transformation, and encodes the gaussian attributes:

$$\mathbf{g} = \{\mathbf{p}, \mathbf{s}, \mathbf{r}, \mathbf{c}, \mathbf{f}\}, \quad \mathbf{g}^T = \mathbf{w}(\mathbf{g}^{T-1}, \mathbf{o}^T), \quad \mathbf{g}_A^T = \text{Align}(\mathbf{g}^{T-1}, \mathbf{M}_{ego}), \tag{8}$$

where $\mathbf{p}, \mathbf{s}, \mathbf{r}, \mathbf{c}, \mathbf{f}$ denote position, scale, rotation, semantic, and temporal features, respectively; Fuction Align aligns the previous gaussian representation $\mathbf{g}^{T-1}$ with the ego-motion matrix $\mathbf{M}_{ego}$ to account for the observer's motion.

Figure 3: The flow of Gaussian Entropy. After Gaussian Entropy is generated in the Entropy Producing process, the entropy space continuously interacts with the environment through Entropy Exchanging. At each interaction, the Entropy Gain from the environment drives the flow of Gaussian Entropy in latent space, ultimately completing the evolution of the scene.

**In-model latent denoising.** Consistent with existing work (Zhou et al., 2025; Ho et al., 2020), we employ an implicit denoising network to model evolutionary entropy flow, decomposing the denoising process into independent modules that enable state transitions for noise queries in a single forward pass. Accordingly, Eq. equation 6 can be written as $p_\theta(S_0 \mid S_I, c) = p(S_I) \prod_{i=\frac{I}{I}}^{I} p_\theta(S_{i-\frac{I}{I}} \mid S_i, c)$, formulating evolutionary entropy flow as an iterative denoising process.

**Gaussian Entropy Flow.** Building on this, we propose the Gaussian Entropy Flow World Model (GaussEFW), which uses an internal denoising network to model scene evolution as a continuous flow of Gaussian Entropy in latent space, providing a more accurate representation of continuous scene changes and dynamic element motion. In contrast, GaussianWorld models scene evolution as discrete gaussian representations movements in physical space using transformer blocks. Initially, dynamic gaussian anchors are perturbed to obtain noisy anchors $g_I$, which are then encoded into latent space via the Entropy Producing mapping to yield Gaussian Entropy $G_I$.

$$g_I^T = \sqrt{\bar{\alpha}} g_A^T \cdot I(g_A^T \in \{g^D\}) + \epsilon \sqrt{1-\bar{\alpha}}, \quad G_I = \mathcal{E}(g_I^T) \tag{9}$$

where $I(\cdot)$ denotes the indicator function selecting dynamic categories, $g^D$ represents the gaussian representation of dynamic objects, and $\epsilon$ is gaussian noise sampled from a standard normal distribution, $\epsilon \sim \mathcal{N}(0, I)$. The operator $\mathcal{E}$ encodes the anchor into the latent space.

Entropy Exchanging is modeled as an iterative prediction over gaussian queries $Q$. In the $i$-th step, $Q_i$ aggregates multi-scale observational features and interacts via self-attention. The resulting $Q_{i-1}$ as Entropy Gain serves as a condition for the current denoising block, guiding the interaction with Gaussian Entropy $G_i$ for subsequent denoising.

$$\text{Attention}_{\text{Denoising}}(Q, G) = \text{Softmax}\left(\frac{q_G k_Q^T}{\sqrt{d}} v_Q\right), \quad G_i' = \text{Attention}_{\text{Denoising}}(Q_{i-1}, G_i) + G_i \tag{10}$$

Building upon the outcome of Entropy Exchanging, Entropy Flow is formulated as an iterative denoising process in the latent space. We perform residual predictions on Gaussian Entropy Flow, which are then used to predict and remove noise, progressively refining the latent representation:

$$G_i'' = G_i' + \text{FFN}(G_i'), \qquad \epsilon = \text{NoisePred}(G_i'') = \text{Linear}(\text{ReLU}(\text{Linear}(G_i''))). \tag{11}$$

The denoised Gaussian Entropy for each block is obtained by subtracting the predicted noise, and after processing all $l$ blocks, the final state is computed by summing denoised contributions:

$$G_{i-\frac{I}{I}} = G_i'' - \epsilon, \quad G_0 = G_I - \sum_{j=1}^{l} \epsilon_j \tag{12}$$

Table 1: 3D occupancy prediction performance on the Surroundocc dataset.

| Method | mIoU (%) | barrier | bicycle | bus | car | cons. veh | motorcycle | pedestrian | traffic cone | trailer | truck | dri. sur | other flat | sidewalk | terrain | manmade | vegetation |
|---|---|---|---|---|---|---|---|---|---|---|---|---|---|---|---|---|---|
| MonoScene (Cao & De Charette, 2022) | 7.31 | 4.03 | 0.35 | 8.00 | 8.04 | 2.90 | 0.28 | 1.16 | 0.67 | 4.01 | 4.35 | 27.72 | 5.20 | 15.13 | 11.29 | 9.03 | 14.86 |
| TPVFormer (Huang et al., 2023) | 17.10 | 15.96 | 5.31 | 23.86 | 27.32 | 9.79 | 8.74 | 7.09 | 5.20 | 10.97 | 19.22 | 38.87 | 21.25 | 24.26 | 23.15 | 11.73 | 20.81 |
| Surroundocc (Huang et al., 2021) | 20.30 | 20.59 | 11.68 | 28.06 | 30.86 | 10.70 | 15.14 | 14.09 | 12.06 | 14.38 | 22.26 | 37.29 | 23.70 | 24.49 | 22.77 | 14.89 | 21.86 |
| OccFormer (Zhang et al., 2023b) | 19.03 | 18.65 | 10.41 | 23.92 | 30.29 | 10.31 | 14.19 | 13.59 | 10.13 | 12.49 | 20.77 | 38.78 | 19.79 | 24.19 | 22.21 | 13.48 | 21.35 |
| BEVFormer (Li et al., 2022) | 16.75 | 14.22 | 6.58 | 23.46 | 28.28 | 8.66 | 10.77 | 6.64 | 4.05 | 11.20 | 17.78 | 37.28 | 18.00 | 22.88 | 22.17 | 13.80 | 22.21 |
| GaussianFormer-B (Huang et al., 2024c) | 19.73 | 19.36 | 13.19 | 26.90 | 29.79 | 10.20 | 15.17 | 12.55 | 9.29 | 12.96 | 21.45 | 39.55 | 23.03 | 25.07 | 23.65 | 12.35 | 21.18 |
| GaussianFormer-T (Huang et al., 2024c) | 20.42 | 20.82 | 12.07 | 26.89 | 30.94 | 10.52 | 16.48 | 13.15 | 10.46 | 12.90 | 21.79 | 41.13 | 24.22 | 26.29 | 24.89 | 12.80 | 21.45 |
| GaussianWorld (Zuo et al., 2024) | 22.13 | 21.38 | 14.12 | 27.71 | 31.84 | 13.66 | 17.43 | 13.66 | 11.46 | 15.09 | 23.94 | 42.98 | 24.86 | 28.84 | 26.74 | 15.69 | 24.74 |
| **GaussEFW -B** | 21.46 | 20.18 | 15.91 | 29.45 | 30.05 | 13.88 | 15.02 | 14.20 | 12.37 | 13.25 | 24.70 | 40.11 | 25.80 | 27.95 | 24.90 | 14.60 | 21.05 |
| **GaussEFW** | **24.22** | 21.72 | 18.68 | 31.66 | 33.86 | 16.81 | 17.87 | 17.44 | 13.96 | 16.14 | 25.17 | 44.06 | 27.73 | 29.42 | 28.18 | 18.08 | 26.88 |

From this process, we obtain Gaussian Entropy $G_0$ at the stable state, and decoding it produces the corresponding evolved target gaussian representation $g_0^T$. For newly completed elements at time $T$, we treat them in the same way as the dynamic elements.

# 4 EXPERIMENTS.

## 4.1 EXPERIMENTAL SETTINGS

**Datasets.** The NuScenes (Caesar et al., 2020) dataset contains 1,000 diverse driving scenarios from Boston and Singapore, split into 700 training, 150 validation, and 150 test sequences. Each 20-second sequence is recorded at 20 Hz, with keyframes at 2 Hz, and includes multi-view RGB images from six cameras. For 3D semantic occupancy prediction, we use dense annotations from SurroundOcc (Wei et al., 2023). The voxel grid spans $[-50, 50]$ m in X and Y, $[-5, 3]$ m in Z, with a resolution of $200 \times 200 \times 16$ (H×W×Z), and each voxel is labeled with one of 18 categories: 16 semantic classes, plus empty and unknown.

**Implementation Details.** We evaluate geometric reconstruction using mIoU across all categories. Input images are set to $900 \times 1600$ and processed with a ResNet101-DCN (He et al., 2015) backbone pre-trained from FCOS3D (Wang et al., 2021), coupled with a Feature Pyramid Network (FPN) (Lin et al., 2017) to extract multi-scale features. The 3D scene is represented by 25,600 Gaussian spheres and iteratively denoised through 6 blocks. Optimization uses AdamW with a learning rate of $4 \times 10^{-4}$ and weight decay of 0.01. The model is trained for 20 epochs on eight A100 GPUs with a batch size of 16, while the encoder and occupancy heads follow prior work (Huang et al., 2024c).

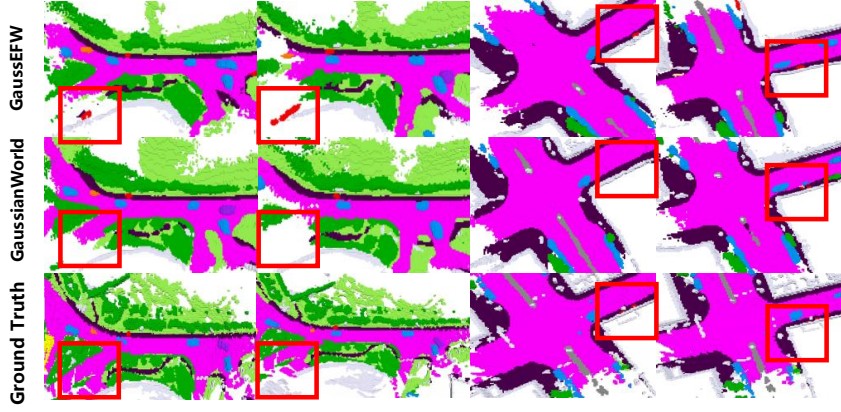

Figure 4: Single-frame visualization comparison. The red boxes highlight that our method performs better on dynamic objects.

## 4.2 TRAINING.

Training GaussianEFW occurs in two phases. In the first, 25,600 semantic Gaussian spheres are sampled from single-frame occupancy grids, perturbed with Gaussian noise, and encoded into latent space to model Gaussian Entropy. Queries interact with multi-scale 2D features to guide the denoising network, producing stable embeddings in a single forward pass and providing basic evolutionary and scene completion capabilities. In the second phase, a streaming strategy (Zuo et al., 2024) feeds scene images sequentially, using predicted 3D Gaussians from the previous frame as the initial state. Training starts with short sequences, gradually increasing length for stability, and is optimized with cross-entropy and Lovász-Softmax losses.

## 4.3 RESULTSANDANALYSIS.

In Table 1, we present results on the nuScenes validation set, comparing our vision-based 3D semantic occupancy prediction method with state-of-the-art approaches using SurroundOcc labels. In the first phase, GaussEFW, trained with a single-frame setup, already outperforms existing methods, including GaussianFormer-B (single frame). In the second phase, GaussEFW further improves upon the single-frame model, the temporal fusion-based GaussianFormer-T (temporal fusion by multi-frames), and the world model GaussianWorld. The underlined values show significant improvements in dynamic object prediction compared to GaussianWorld. GaussEFW boosts mIoU by 2.8 over the single-frame baseline and by 2.1 over GaussianWorld for streaming occupancy prediction. These results demonstrate the effectiveness of our evolutionary entropy flow-based world model, surpassing both traditional temporal fusion methods and existing scene evolution-based models. Visualizations for single-frame and streaming tasks are shown in Figures 4 and 5.

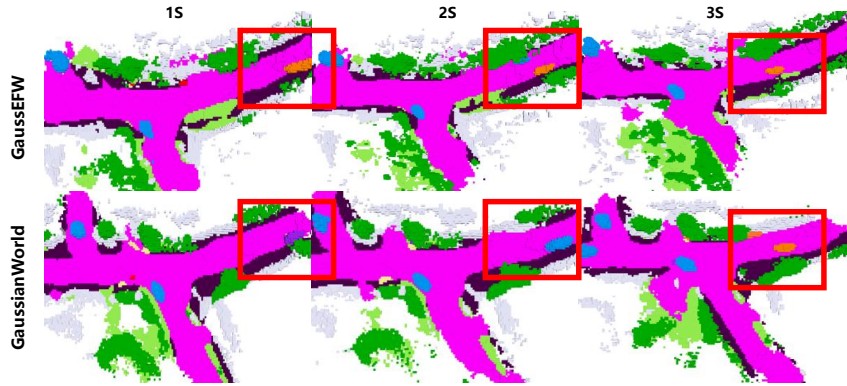

Figure 5: Streaming visualization comparison, the red boxes highlight that our method shows better performance and consistency for dynamic objects.

## 4.4 ABLATION STUDY.

**Ablation on Category-Wise Noise.** As shown in Table 2, adding noise to all categories in streaming occupancy prediction significantly reduces performance compared to adding noise only to dynamic categories. Because uniform entropy modeling weakens the high-frequency features of dynamic elements and introduces noise into stable regions, reducing the precision of static representations.

| Category-Wise | mIoU | barrier | bicycle | bus | car | const. veh. | motorcycle | pedestrian | traffic cone | trailer | truck | drive. suf. | other flat | sidewalk | terrain | manmade | vegetation |
|---|---|---|---|---|---|---|---|---|---|---|---|---|---|---|---|---|---|
| Dynamic Categories | 24.22 | 21.72 | 18.68 | 31.66 | 33.86 | 16.81 | 17.87 | 17.44 | 13.96 | 16.14 | 25.17 | 44.06 | 27.73 | 29.42 | 28.18 | 18.08 | 26.88 |
| All Categories | 21.48 | 18.36 | 15.42 | 25.83 | 29.71 | 13.95 | 13.23 | 15.37 | 8.41 | 17.78 | 23.62 | 42.83 | 23.59 | 23.47 | 28.35 | 17.52 | 26.47 |
| diff | -2.74 | -5.36 | -1.26 | -3.83 | -4.15 | -2.86 | -6.64 | -2.07 | -5.55 | +1.64 | -1.55 | -1.23 | -4.14 | -5.95 | +0.17 | -0.56 | -0.41 |

Table 2: Ablation study of Category-Wise Noise.

Table 3: Ablation Study on Entropy Producing and Exchanging.

| | Entropy Producing | Entropy Exchange | mIoU |
|---|---|---|---|
| a | × | ✓ | 19.92 |
| b | ✓ | × | 17.18 |
| c | ✓ | ✓ | **24.22** |

Table 4: Ablation on Denoising Block Number.

| Block Number | mIoU(Streaming) |
|---|---|
| 2 | 15.32 |
| 3 | 15.28 |
| 4 | 18.81 |
| 5 | 22.70 |
| **6** | **24.22** |
| 7 | 23.85 |

Table 5: Ablation Study on Streaming Length.

| Length | 1 | 5 | 10 | 15 | 20 | 25 | 30 | 38 |
|---|---|---|---|---|---|---|---|---|
| 4 | 16.22 | 17.71 | 17.12 | 18.04 | 18.22 | **18.79** | 16.32 | 15.70 |
| 6 | 19.35 | 21.56 | 23.67 | **24.22** | 23.58 | 22.31 | 21.44 | 20.72 |
| 8 | 18.47 | 18.81 | **20.74** | 20.04 | 19.98 | 17.96 | 16.92 | 16.6 |

**Ablation Study on Streaming Length.** We investigate the impact of streaming length on model performance with different denoising block configurations (Table 5). Performance improves as the number of frames increases, with the evolutionary entropy flow framework stabilizing Gaussian Entropy. However, with 6 denoising blocks, performance declines after approximately 15 frames, likely due to excessive entropy accumulation that disrupts denoising. Adding more denoising blocks boosts representational capacity but lowers the critical frame threshold, making the model more sensitive to entropy accumulation, potentially affecting stability in long-sequence scenarios.

**Ablation Study on the Number of Denoising Blocks.** We study the impact of denoising block number on performance in streaming tasks. Using forward-only blocks, performance is low at steps two and three, improves by the fourth, peaks at the sixth (Table 4), and declines beyond the seventh. Too few blocks limit Gaussian Entropy flow, reducing the use of environmental observations, while too many blocks introduce excessive information, disrupt entropy flow, and may cause overfitting, degrading performance.

**Ablation Study on Entropy Producing and Exchanging** We conducted an ablation study on Entropy Producing (EP) and Entropy Exchanging (EE) in streaming tasks as tabel 3 shown. Three groups were tested: A (EP+, EE+) uses noisy denoising blocks and retains EE; B (EP+, EE-) uses noisy denoising blocks with deformable attention and 2D features, removing EE; C (EP-, EE+) uses noise-free Transformer blocks while retaining EE. The results show that EP provides the stochastic driving force, EE guides entropy along stable trajectories, and disabling either or both reduces performance, confirming their complementary roles in modeling dynamic behaviors.

## 5 CONCLUSION

We propose the evolutionary entropy flow framework for continuous 3D scene prediction, modeling dynamic foreground motion as evolutionary entropy flow. Based on this framework, GaussEFW leverages Gaussian latent representations and an implicit denoising network to capture fine-grained temporal evolution, reducing error accumulation from discrete encoding-decoding processes. By integrating observational information through attention mechanisms, GaussEFW accurately predicts dynamic element attributes and coherent scene evolution. Experiments on the nuScenes dataset validate the effectiveness of GaussEFW, demonstrating superior performance in dynamic element prediction and overall occupancy accuracy.

## 6 ETHICS STATEMENT

This research adheres to the ICLR Code of Ethics. No human or animal experimentation was involved, and all datasets were obtained in compliance with privacy guidelines. Efforts were made to eliminate biases and prevent discrimination. No personally identifiable information was used, and no privacy or security risks were posed. We are committed to transparency and ethical integrity throughout the research process.

## 7 LLM USAGE

This study used a Large Language Model (LLM) to assist with language, readability, and clarity. The LLM did not contribute to ideation, methodology, or experimental design. All research concepts and analyses were independently developed by the authors. The LLM's role was solely linguistic, with no involvement in scientific content. The authors take full responsibility for the manuscript and ensure the LLM-generated text adheres to ethical guidelines, avoiding plagiarism.

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

## A    DETAILS OF ENTROPY EXCHANGING

Entropy Exchanging iteratively generates and refines entropy gains $Q_{i-1}$, continuously guiding the update of Gaussian entropies $G_i$. In this process, each Gaussian Entropy leverages gaussian queries to extract information from the environmental observation $o^T$ for generating and optimizing Entropy Gains. The overall Entropy Exchanging process consists of two stages: generation of Entropy Gains and refinement of Entropy Gains.

During the Entropy Gain generation stage, for each Gaussian Entropy $G_i$, a set of reference points is generated:

$$R = \{m + \Delta m_j\}_{j=1}^R,$$

where the offsets $\Delta m_j$ are sampled according to the Gaussian covariance to reflect the spatial extent of the Gaussian Entropy. Each reference point $R_j$ is projected onto the image planes of multiple views via camera intrinsics $K$ and extrinsics $T$. Then, the next-state Entropy Gain is computed from the current Entropy Gain $Q_i$ using the Entropy Gain generation mapping:

$$Q_{i-1} = \mathcal{E}_E(G_i, Q_i, o^T),$$

which can be implemented using deformable attention as:

$$Q_{i-1} = \frac{1}{N} \sum_{n=1}^N \sum_{j=1}^R DA(Q_i, \pi(R_j; T, K), o_n),$$

where $DA(\cdot)$ denotes the deformable attention operator. The Entropy Gain $Q_i$ carries high-dimensional information and serves as the core state for extracting and fusing environmental features in the Entropy exchanging process.

In the Entropy Gain refinement stage, the computed Entropy Gain $Q_{i-1}$ is iteratively fused and interacted through sparse convolution among gaussian queries, thereby refining the Entropy Gain and providing guidance for updating the Gaussian Entropy $G_i$ in the next iteration.

## B    DETAILS OF ENTROPY FLOWING

The evolutionary entropy is defined as

$$S(p) = - \int p(x) \log p(x) \, dx. \tag{13}$$

To predict the target state $r_0$, we use a probabilistic model $p_\theta(r_0 \mid r_I, c)$, where $\theta$ denotes the neural network parameters. The optimization goal is to maximize the predicted probability of $r_0$, aggregating evolutionary feature information for iterative refinement:

$$\text{maximize}\big(p_\theta(r_0 \mid r_I, c)\big). \tag{14}$$

According to the Markov principle, the probability of the target state can be factorized as the product of the initial state probability and a series of conditional probabilities:

$$p_\theta(r_0 \mid r_I, c) = p(r_I) \prod_{i=1}^I p_\theta(r_{i-1} \mid r_i, c_i), \tag{15}$$

where $r_I$ is the initial state, and $c_i$ represents the context at each step.

Taking the logarithm yields

$$\log p_\theta(r_0 \mid r_I, c) = \log p(r_I) + \sum_{i=1}^I \log p_\theta(r_{i-1} \mid r_i, c_i). \tag{16}$$

The evolutionary entropy of the stable state $S_0$ is defined as

$$S_0 = - \int p_\theta(r_0 \mid r_I, c) \log p_\theta(r_0 \mid r_I, c) \, dr_0. \tag{17}$$

Substituting the Markov factorization, it can be decomposed into the initial state entropy and the sum of conditional entropies:

$$S_0 = - \int p(r_I) \log p(r_I) \, dr_I - \sum_{i=1}^{I} \int p_\theta(r_{i-1} \mid r_i, c_i) \log p_\theta(r_{i-1} \mid r_i, c_i) \, dr_{i-1}. \qquad (18)$$

The initial state entropy is

$$S_I = - \int p(r_I) \log p(r_I) \, dr_I, \qquad (19)$$

so the stable-state entropy can be expressed as

$$S_0 = S_I + \sum_{i=1}^{I} S_i, \qquad (20)$$

where

$$S_i = - \int p_\theta(r_{i-1} \mid r_i, c_i) \log p_\theta(r_{i-1} \mid r_i, c_i) \, dr_{i-1}. \qquad (21)$$

Since the entropy flow $S_i'$ can be interpreted as the reduction in evolutionary entropy from the initial state to the stable state, we define

$$S_i' = -S_i < 0. \qquad (22)$$

Therefore, the stable-state evolutionary entropy can be written in the subtractive form:

$$S_0 = S_I - \sum_{i=1}^{I} S_i', \qquad (23)$$

By predicting each step's evolutionary entropy flow $S_i'$, the entropy monotonically decreases along the flow, eventually converging to the stable state $r_0$.

## C  MORE EVALUATIONS

**Ablation Study on Noise Corruption Timestamps.**  Table 6 reports the performance of the streaming and single-frame models at different noise corruption timestamps $\tau$. The single-frame model performs best at $\tau = 1400$, while the streaming model performs best at $\tau = 800$. We observe that the single-frame model requires noise and denoising across all categories to capture the evolution of all elements, thus needing more noise. As $\tau$ increases, performance improves, but after a peak, it starts to degrade. Because more time steps allow the model to generate evolutionary entropy and approach the global optimum. Too few steps limit dynamic element modeling, while too many steps overload the denoising blocks, preventing entropy from stabilizing and causing performance drops.

**Ablation Study on Entropy Exchanging**  We investigate the role of Entropy Gains generated by the environment's self-evolution during the Entropy exchanging process as 8 shown. In the experiment, Cross-Attention in the denoising blocks is replaced with Deformable Attention, and Gaussian queries are replaced with 2D backbone features. This removes the reliance on Entropy exchanging, directly guiding denoising through 2D observations. The results show that Entropy exchanging is crucial for perceiving dynamic elements. Without it, denoising relies only on static features, lacking the environment's iterative self-refinement, which impairs the capture of high-frequency behaviors and degrades overall performance.

**Ablation Study on Denoising Blocks**  We investigated the impact of denoising blocks on scene evolution (Table 7), dividing the experiments into two groups: (a) denoising blocks and (b) Transformer blocks. In group (a), noise is added to enable scene evolution via evolutionary entropy flow. In group (b), no noise is used, and Gaussian queries update anchors with attention and latent offset features. The results show a significant decline in dynamic element performance, highlighting that relying solely on observational features and anchor embeddings is insufficient for effective scene evolution. Without noise in group (b), Gaussian entropy is not generated, limiting denoising and updates, which hinders capturing dynamic scene changes.

**Visualization.**  We performed additional visualizations for the streaming occupancy prediction, as shown in Figure 6. We can observe that our method performs better in scenes with more dynamic elements and in larger spatial areas of dynamic objects.

Table 6: Ablation Study on Noise Corruption Timestamps.

| corruption timestamp | 200 | 400 | 600 | 800 | 1000 | 1200 | 1400 | 1600 | 1800 |
|---|---|---|---|---|---|---|---|---|---|
| Streaming | 14.25 | 15.62 | 22.31 | **24.22** | 24.01 | 21.82 | 16.94 | 13.86 | 9.20 |
| Single frame | 8.85 | 8.66 | 11.32 | 16.37 | 20.70 | 21.25 | **21.46** | 20.08 | 18.35 |

Table 7: Ablation Study on Denoising Blocks: Comparing the Impact of Denoising and Transformer Blocks on Scene Evolution Across Dynamic Object Categories.

| Group | Bicycle | Bus | Car | Motorcycle | Pedestrian |
|---|---|---|---|---|---|
| (a) Denoising Blocks | 18.68 | 31.66 | 33.86 | 17.87 | 17.44 |
| (b) Transformer Blocks | 16.11 | 27.89 | 31.55 | 14.12 | 15.38 |

Table 8: Ablation Study on Entropy exchanging for Dynamic Object Performance.

| Condition | Bicycle | Bus | Car | Motorcycle | Pedestrian |
|---|---|---|---|---|---|
| With Entropy exchanging | 18.68 | 31.66 | 33.86 | 17.87 | 17.44 |
| Without Entropy exchanging | 12.72 | 23.16 | 28.53 | 14.78 | 14.89 |

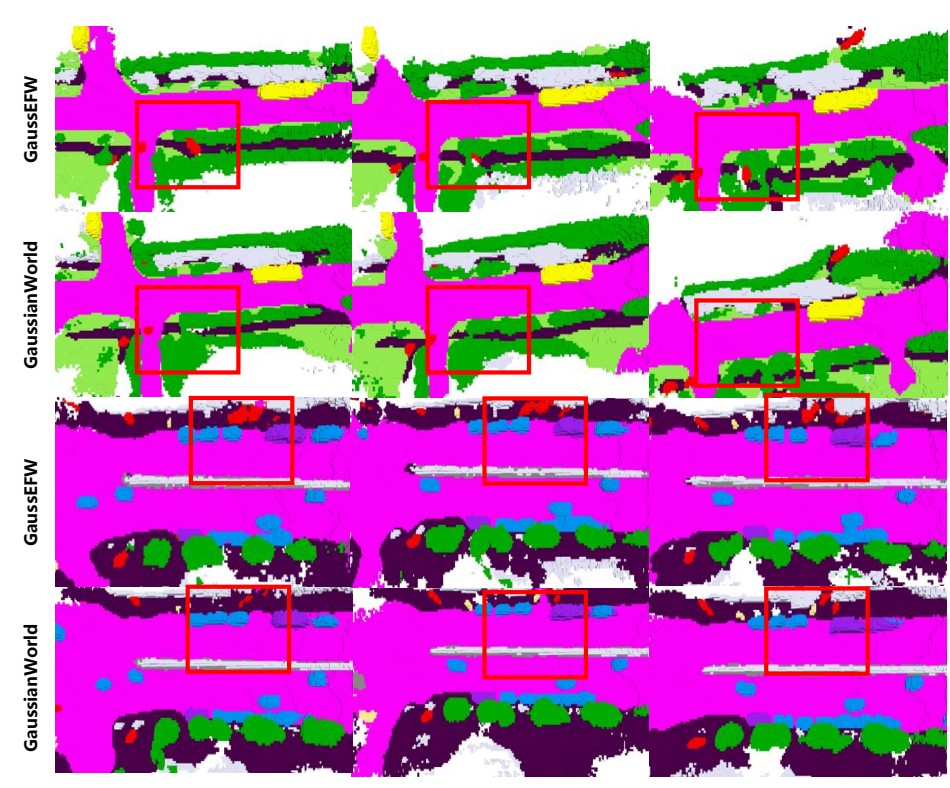

Figure 6: Additional visualizations of streaming occupancy prediction, showing improved performance in dynamic scenes.

