# OpenReview forum: "Gaussian Entropy Flow World Model for Streaming 3D Occupancy Predition"
_ICLR.cc/2026/Conference — ICLR 2026 Conference Withdrawn Submission_

### Official Review · Reviewer_JvHV · 2025-10-31

**Soundness:** 2
**Presentation:** 2
**Contribution:** 2
**Rating:** 4
**Confidence:** 4

**Summary:**

This paper presents the Gaussian Entropy Flow World Model (GaussEFW) for streaming 3D occupancy prediction in dynamic scenes. Building on principles from non-equilibrium thermodynamics, the authors propose modeling the evolution of dynamic scene elements as a continuous flow of Gaussian entropy within a latent space, instead of relying on discrete refinement across multiple encoding-decoding steps. The framework introduces three processes—Entropy Producing, Entropy Exchanging, and Entropy Flowing—to track and refine dynamic elements in a unified way. Experimental results on nuScenes with SurroundOcc labels demonstrate improved dynamic occupancy prediction and overall mIoU when compared against state-of-the-art methods, including GaussianWorld.

**Strengths:**

-  The paper’s central idea of representing dynamic scene evolution as a continuous Gaussian entropy flow in latent space (as opposed to repeated discrete refinements) is conceptually interesting and aims to address error accumulation—an important shortcoming of current world models. This is visually well-motivated in Figure 1, where the trajectories and error types associated with existing models (discrete, cumulative refinement) are contrasted against the proposed continuous entropy flow.
- The illustration of the GaussEFW architecture in Figure 2 (Page 4) and the associated algorithmic breakdown provide readers with a clear, stepwise perspective on the denoising network, entropy production, and block interactions. This is supported by detailed equations specifying the entropy production, flow, and denoising processes (Equations on Pages 5-6).
- Multiple ablation studies are presented (Tables 2, 5, 6, 7, 8; Pages 8-15), dissecting the impact of the entropy-producing/exchange mechanisms, block configurations, category-wise noise, streaming length, and timestamp corruption. This demonstrates a commitment to experimental rigor.

**Weaknesses:**

- The performance of occupancy forecasting is relatively low compared to previous state-of-the-art (SOTA) occupancy prediction works, such as UniScene (mIoU 31.76). The authors are encouraged to include comparisons with these methods to provide a more complete benchmark.

- It would be valuable to investigate longer-term predictions and analyze how errors accumulate over time. Currently, there is limited discussion of failure cases or scenarios where the model underperforms. For example, although Figures 4, 5, and 6 highlight improvements, the paper lacks a systematic error or failure analysis. Moreover, the quantitative limitations of GaussEFW under very long streaming sequences are not thoroughly characterized, aside from a brief mention in Table 5.

- In several ablation studies (e.g., Table 2 and Table 7), results are presented without sufficient interpretation or explanation. For instance, why does applying noise only to dynamic categories lead to sharp performance changes? A deeper investigation into these observed effects would strengthen the paper’s analysis.

- While speed and error accumulation are central motivations of the work, the paper does not include benchmarking results on inference speed, memory usage, or computational efficiency compared to recent fast-splatting models. Including these comparisons would make the contributions more convincing and practically relevant.

**Questions:**

Please refer to the weakness above.

---

> ### Author Response · Authors · 2025-11-17
>
> We thank Reviewer JvHV for their insightful and positive review.
>
> 1. Although our method is related to world-model approaches, our task is streaming occupancy prediction, whereas UniScene focuses on 4D occupancy generation. Our primary goal is to demonstrate improvements over prior work specifically within the streaming occupancy prediction setting, thereby validating the effectiveness of our method for this task.
>
> 2. Our analysis shows that, over time, we alleviate the accumulated errors of dynamic objects caused by the repeated encoding–decoding process in GaussianWorld. However, as discussed in GaussianWorld, another performance bottleneck arises from the quality of the ground-truth annotations. Existing 3D occupancy labels are obtained by fusing multiple LiDAR frames, which makes annotations near scene boundaries relatively sparse. This sparsity affects long-horizon prediction, particularly around boundary regions, and also explains the degradation observed when extending the sequence length in our analysis.
>
> 3. As stated in the paper, adding noise to all categories significantly degrades performance in streaming occupancy prediction. Injecting uniform noise across the entire scene leads to homogeneous entropy modeling, which suppresses the high-frequency features of dynamic elements. In contrast, injecting noise only into dynamic objects better models and highlights their high-frequency behaviors.
>
> 4. Comparison and analysis of inference efficiency and memory consumption:
>
> | Temporal Model   | Number of Historical Input | Latency | Memory | mIoU  |
> | ------------------ | ---------------------------- | --------- | -------- | ------- |
> | 3DGaussianFusion | 3                          | 379ms   | 9993M  | 20.24 |
> | GaussianWorld    | 1                          | 228ms   | 7030M  | 21.87 |
> | GaussEFW         | 1                          | 242ms   | 7416M  | 24.22 |
>
> * From this, although our method introduces the denoising strategy under the evolution-entropy paradigm (which is not a generative task), the latency and memory consumption do not increase significantly. Under similar computational budgets to GaussianWorld, we achieve better performance. We attribute this to the effectiveness of the evolution-entropy paradigm and the simplicity and efficiency of the denoising block design.

---

### Official Review · Reviewer_t2wr · 2025-10-31

**Soundness:** 4
**Presentation:** 3
**Contribution:** 2
**Rating:** 6
**Confidence:** 4

**Summary:**

The paper introduces an Evolutionary Entropy Flow framework for modeling the continuous evolution of dynamic 3D elements in 3D occupancy scenes. The proposed GaussEFW uses Gaussian latent representations and an implicit denoising network to model continuous scene evolution. It reformulates dynamic scene prediction as a Gaussian Entropy Flow process in latent space, contrasting with previous discrete refinement methods. Experiments on the nuScenes dataset with SurroundOcc annotations show stronger performance in dynamic element prediction and higher overall performance.

**Strengths:**

1. The paper presents an interesting and imaginative approach that builds a conceptual bridge between physics-inspired entropy flow and continuous world modeling. The idea of treating dynamic motion as an entropy flow in latent Gaussian space is conceptually novel.
2. This paper is essentially well-written and easy to understand.

**Weaknesses:**

1. While the proposed method is novel, its significance is limited to a technical contribution. For example, only nuScenes is used for quantitative results. Tests on other datasets (e.g., SemanticKITTI) would better demonstrate generalization to diverse motion scenarios.
2. Regarding the statement "Through these mechanisms, GaussEFW provides an efficient and robust approach for continuous 3D scene prediction...," the paper lacks a clear analysis of the computational efficiency and scalability of GaussEFW compared to previous works such as GaussianWorld.
3. Evaluation against non-Gaussian temporal 3D methods (e.g., UniScene) is not discussed.
4. Minor: Several mathematical formulations (e.g., Eq. 7) are introduced without sufficient explanation or intuition, which makes them difficult to understand. The capitalization of some letters (e.g., Per, Trans, and Fuse in Eq. 2) should be formatted consistently.

**Questions:**

1. I wonder if it is possible to generate the trajectory of the ego vehicle as the future occupancy is generated. What's the planning performance compared to related works such as UniAD, GenAD, etc.?
2. Is it possible to visualize more seconds in Figure 5? I'm curious how far the world model can see.
3. I think in Figure 4, the results of GaussianWorld are smoother and better, especially the foreground, like the road.
4. Does the continuous flow modeling increase inference latency?

---

> ### Author Response · Authors · 2025-11-17
>
> We thank Reviewer t2wr for their insightful and positive review.
> 1. Regarding the evaluation:
>
> * Our baseline, GaussWorld, is evaluated under the SurroundOcc benchmark. To fairly demonstrate our improvements over it, we follow the same evaluation protocol. Our contribution enhances GaussWorld within its streaming occupancy prediction framework, improving baseline performance while fully retaining the advantages of streaming prediction.
>
> 2. Comparison and analysis of inference efficiency and memory usage:
>
> | Temporal Model   | Number of Historical Input | Latency | Memory | mIoU  |
> | ------------------ | ---------------------------- | --------- | -------- | ------- |
> | 3DGaussianFusion | 3                          | 379ms   | 9993M  | 20.24 |
> | GaussianWorld    | 1                          | 228ms   | 7030M  | 21.87 |
> | GaussEFW         | 1                          | 242ms   | 7416M  | 24.22 |
>
> * From this, although our method introduces the denoising strategy under the evolution-entropy paradigm (which is not a generative task), the latency and memory consumption do not increase significantly. Under similar computational budgets to GaussianWorld, we achieve better performance. We attribute this to the effectiveness of the evolution-entropy paradigm and the simplicity and efficiency of the denoising block design.
>
> 3. Although related to world-model approaches, our task is streaming occupancy prediction, whereas UniScene focuses on 4D occupancy generation. Our primary goal is to demonstrate improvements over prior work under the streaming occupancy prediction setting, validating the effectiveness of our approach.
>
> 4. Regarding formulas and derivations:
>
> *  The appendix includes the complete derivation of Equation (7). We acknowledge the formatting issues and will refine them accordingly.
>
> 5. Responses to your questions:
>
> *  Our task is streaming 3D occupancy perception rather than an end-to-end pipeline, which is fundamentally different from generative methods such as UniScene.
> *  Compared with static objects, our method particularly improves the perception quality of dynamic objects, where the advantages of our approach become even more pronounced.

---

### Official Review · Reviewer_XD9K · 2025-11-01

**Soundness:** 2
**Presentation:** 2
**Contribution:** 2
**Rating:** 2
**Confidence:** 5

**Summary:**

This paper proposes GaussEFW, an approach to 3D occupancy prediction that models continuous scene evolution through an evolutionary entropy flow mechanism inspired by nonequilibrium thermodynamics. GaussEFW represents temporal dynamics as a single continuous Gaussian Entropy Flow in latent space. The method introduces three key processes, Entropy Producing, Entropy Exchanging, and Entropy Flowing to capture the motion of dynamic elements. Experiments on the SurroundOcc dataset demonstrate improvements over GaussianWorld and GaussianFormer variants.

**Strengths:**

1. The idea of implicit temporal fusion in latent Gaussian space is intersting.

2. The method shows empirical improvement on the SurroundOcc dataset over previous Gaussian-based approaches.

3. The attempt to draw an analogy with nonequilibrium thermodynamics is intellectually creative.

**Weaknesses:**

1. The reviewer’s main concern is that the use of nonequilibrium thermodynamics as an explanatory framework is unconvincing. It is difficult to see a direct connection between the 3D occupancy prediction task and the notion of entropy. The core technical contribution, i.e, modeling the dynamic evolution of Gaussian kernels in latent space, can be fully developed without invoking thermodynamic principles. Framing the work under this lens makes the conceptual motivation unclear and obscures the underlying technical ideas.

2. Fig. 1 (b) does not clearly illustrate what changes are driven by the proposed entropy flow. The visual explanation fails to make evident the mechanism or benefit of this flow, leaving the reader uncertain about the core message that Fig. 1 is meant to convey.

3. Eqs. (2) and (3) appear to describe similar processes using different notations, making it unclear what distinct roles they play. The use of the term world model to denote the network w is ambiguous, as “world model” can refer to multiple subfields and paradigms. While it seems that the authors intend to describe a temporal modeling process akin to an RNN-based fusion scheme, more precise terminology is needed to eliminate confusion and to clarify what is genuinely novel.

4. The paper does not sufficiently describe the 3D occupancy prediction task itself—particularly the final loss function or optimization objectives. Without these details, the method section remains incomplete, and readers who are not already familiar with this research area will struggle to reproduce or properly interpret the approach.

5. No results are provided on computational efficiency, such as inference FPS or parameter count. Since the proposed temporal fusion mechanism increases model complexity, efficiency is a crucial consideration in 3D occupancy prediction. Without reporting inference efficiency or comparing with baselines, it is unclear whether the performance gain stems from a more effective design or simply from added parameters and computation.

6. The left-hand example in Figure 4 shows noticeable discrepancies between the predicted occupancy and the ground truth. However, it is unclear what the color coding represents or what qualitative aspect is being highlighted. Including the corresponding camera image as reference and annotating which colors correspond to which object categories would make the qualitative analysis more interpretable.

7. The experimental validation is limited to the SurroundOcc dataset. However, the mainstream benchmark for nuScenes-based occupancy prediction is Occ3D, and evaluation on this dataset is necessary for fair comparison with SOTA methods such as COTR [1], FB-Occ [2], and GDFusion [3]. Furthermore, results on additional datasets (e.g., SemanticKITTI, Waymo-Occ) are recommended to demonstrate generalization. Reporting the standard IoU metrics used by prior works would also make the comparison clearer and more credible.

8. GDFusion [3] achieves 25.5 mIoU on the SurroundOcc benchmark, yet it is not included in the comparison table. This omission weakens the empirical claims, as GDFusion represents a strong baseline closely related to the proposed approach.

9. The paper’s use of the term world model seems conceptually narrow. The proposed method focuses on temporal fusion for occupancy prediction, rather than the broader predictive or generative modeling typically associated with world models. Prior works in temporal fusion, such as StreamPETR or GDFusion, also address dynamic scene evolution; hence, the authors should clarify what distinctive aspects of their approach justify the “world model” label.

Minor：

10. L-81: There is a missing space after the period in “process.Collectively.”

11. The subsection title at L-161 lacks a terminating period.

12. The titles of Sections 1, 2, 3, and 5 do not end with a period, whereas Section 4 and several of its subsections do. The punctuation style should be standardized across all section and subsection titles.

13. It is unclear whether the columns in Tab. 5 correspond to the number of blocks or another architectural component. This should be explicitly stated either in the table header or in its caption.

14. Several recent works specifically designed for temporal fusion in occupancy prediction are not discussed, including GDFusion [3] and CVT-Occ [4]. Adding a comparison or a discussion of their relationship to the proposed method would provide better context and strengthen the paper’s positioning.

[1] COTR: Compact Occupancy Transformer for Vision-Based 3D Occupancy Prediction, CVPR 2024

[2] FB-Occ: 3D Occupancy Prediction Based on Forward–Backward View Transformation

[3] Rethinking Temporal Fusion with a Unified Gradient Descent View for 3D Semantic Occupancy Prediction, CVPR 2025

[4] CVT-Occ: Cost Volume Temporal Fusion for 3D Occupancy Prediction, ECCV 2024

**Questions:**

1. In L-210, the phrase “While this method performs well in static scenes,” is unclear. What specific method does “this method” refer to? The preceding section only defines a general temporal modeling paradigm but does not describe any particular implementation. Please clarify which approach is being referenced here.
2. Could the authors elaborate on how E_P in Eq. (5) is implemented in practice, and how it interacts with or extends Eq. (4)? A more explicit description of this module’s design and its computational role within the overall entropy flow framework would greatly improve clarity.

---

> ### Author Response · Authors · 2025-11-17
>
> We thank Reviewer XD9K for their insightful and positive review.
> 1. We believe the introduction of non-equilibrium thermodynamics is meaningful because our goal is to reduce the loss incurred when Gaussian primitives move between encoding and decoding, and we hope to enable an implicit way for them to move within the latent space. To allow their positions to update in latent space, we reformulate the process of predicting position updates in physical space into a process of predicting latent noise and denoising it. The concept of evolution entropy, as well as the associated processes of entropy production and entropy flow, provides a natural explanation for such scene evolution. Entropy production introduces the idea of entropy flow—i.e., injecting noise into the scene representation to construct an evolution gradient—while entropy flow guides the evolution entropy to move toward directions informed by environmental observations.
>
> * Evolution Entropy: It describes how the system’s entropy evolves as a whole during a process. This total change results from both internal processes and boundary interactions, representing the dynamic trend of the system’s disorder. In our paper, it is defined as a high-dimensional latent state carrying the motion information of dynamic scene elements, serving as the core carrier of continuous scene evolution.
>
> * Entropy Production: Entropy production specifically describes the rate at which entropy is created by irreversible processes internal to the system. It directly measures the system’s irreversibility and represents the dissipation or degradation of energy quality—the intrinsic source driving the system toward disorder. In our context, it refers to injecting dynamic elements’ states into the “entropy space” by introducing controlled stochastic perturbations, providing an initial uncertain driving force for scene evolution.
>
> * Entropy Flow: Entropy flow refers to the rate of entropy crossing the system boundary, associated with energy exchange with the environment. It represents entropy exchanged between the system and the external world. It describes the process through which evolution entropy flows continuously and smoothly in the latent space—from one state to another—guided by environmental observation information (entropy exchange), eventually converging to a stable scene state.
>
> 2. Compared to GaussWorld, we introduce the concept of evolution entropy:
>
> * Single Space: Instead of repeatedly encoding and decoding between the feature space and physical space—which leads to coding loss—we let the scene evolve implicitly within a single feature space.
>
> * Continuous Flow: Instead of decoding back to physical space for incremental updates (which introduces motion decoding errors), we perform updates directly through denoising in the latent space.
>
> * In Fig.1(b), compared to (a), we unify the multiple Encode→Refine cycles (black boxes) into a single encoding step followed by decoding.
>
> 3. Explanations of the formulas and the adopted world-model paradigm:
>
> * For Equation (2), we follow the perception–alignment–fusion paradigm, where p is the pose of the observation and a is the temporally aligned feature.
>
> * For Equation (3), we adopt the paradigm proposed by GaussWorld: using the evolution between the previous and current timestep to replace the temporal-fusion scheme mentioned by the reviewers. This is not a generative world model. As in GaussWorld, our formulation of a streaming occupancy-prediction world model is: rather than re-perceiving the scene from scratch at each sample time for every previous timestep (a costly and prior-agnostic approach), the scene representation at the previous state evolves to the next state using the current observation as a prompt and leveraging the model’s own priors. We believe this aligns with the essential meaning of a world model.
>
> 4. Comparison and analysis of inference efficiency and memory usage:
>
> | Temporal Model   | Number of Historical Input | Latency | Memory | mIoU  |
> | ------------------ | ---------------------------- | --------- | -------- | ------- |
> | 3DGaussianFusion | 3                          | 379ms   | 9993M  | 20.24 |
> | GaussianWorld    | 1                          | 228ms   | 7030M  | 21.87 |
> | GaussEFW         | 1                          | 242ms   | 7416M  | 24.22 |
>
> * From this, although our method introduces the denoising strategy under the evolution-entropy paradigm (which is not a generative task), the latency and memory consumption do not increase significantly. Under similar computational budgets to GaussianWorld, we achieve better performance. We attribute this to the effectiveness of the evolution-entropy paradigm and the simplicity and efficiency of the denoising block design.

---

> > ### Author Response · Authors · 2025-11-17
> >
> > 5. Regarding SurroundOcc evaluation:
> >
> > * Our baseline, GaussWorld, is evaluated using SurroundOcc; therefore, to demonstrate improvements over it, we follow the same evaluation protocol.
> >
> > * Our contribution aims to improve GaussWorld within its streaming occupancy-prediction framework, enhancing baseline performance while retaining the advantages of streaming prediction.
> >
> > 6. Comparison with GDFusion:
> >
> > * Although GDFusion’s ALOcc-GF variant achieves 25.5 mIoU on the SurroundOcc benchmark, its memory consumption is 11857 MB. Under similar mIoU performance, our method is significantly more memory-efficient.
> >
> > 7. Regarding the questions:
> >
> > * Question 1:
> > “This method” refers to the prior streaming occupancy-prediction world model proposed in GaussWorld. Its stronger performance on static scenes is relative to conventional perception–alignment–fusion temporal modeling approaches.
> >
> > * Question 2:
> > (1) Since we express the implicit movement of Gaussians as an indirect denoising process acting on Gaussian evolution entropy, Equation (4) provides the mathematical formulation for introducing evolution entropy and prepares for the subsequent entropy-flow process.
> > (2) Equation (5) implements this mathematical formulation: Noise adds perturbation to the Gaussians in physical space at the previous stage, and the subsequent E_P encodes the noised Gaussians into feature space.

---

> > > ### Comment · Reviewer_XD9K · 2025-11-18
> > >
> > > Thank you for your response to. I appreciate the clarifications provided. However, several of my core concerns remain, and I would like to offer further feedback on your rebuttal.
> > >
> > > 1. Regarding non-equilibrium thermodynamics :
> > >
> > > I understand the authors' motivation to model Gaussian updates within the latent space and the appeal of using non-equilibrium thermodynamics as an explanatory framework. But I think the process of updating Gaussian features can be more clearly and sufficiently described using established machine learning terminology, such as "latent space refinement" or "residual updates." The link between the network feature, which you denote as entropy G, and the physical meaning of entropy or its mathematical definition in Eq. (4) remains tenuous. Simply labeling a feature vector as "entropy" without a rigorous and well-motivated connection feels forced and, in my view, obscures the technical contribution rather than clarifying it.
> > >
> > > 2. Regarding "World Model" Terminology and Formulation:
> > >
> > > Regarding the formulation in Eqs. (2) and (3), I find the distinction presented to be misleading. Fundamentally, both formulations can be used to describe nearly all temporal fusion paradigms, including traditional multi-frame methods (e.g., SoloFusion, FB-Occ) and streaming approaches (e.g., StreamPETR, GaussianWorld, GDFusion). The paradigm described in Eq. (2) can easily be reframed to fit the structure of Eq. (3), which makes the separation of these two formulas confusing rather than clarifying.
> > >
> > > I understand that the key differentiators between multi-frame and streaming approaches (which this paper labels as "world models") lie not in the high-level formula, but in the specifics: namely, the representation of the historical state and the design of the update function.
> > >
> > > This leads to my broader concern about the use of the term "world model." While I see the conceptual resemblance, the term is applied too broadly here. If any method that evolves a hidden state based on new observations is classified as a "world model," then by that logic, prior streaming methods like StreamPETR and GDFusion, and indeed all RNNs, could claim the same label. This dilution of the term's meaning creates practical confusion. For instance, it might lead other reviewers to incorrectly associate this work with occupancy forecasting methods like UniScene, which operate in a distinct subfield. The paper's contribution would be communicated more clearly and accurately by positioning it precisely within the well-established context of streaming temporal fusion, rather than claiming the broader and more ambiguous "world model" paradigm.
> > >
> > > 3. Regarding Experimental Evaluation:
> > >
> > > While I understand the desire to maintain consistency with the direct baseline (GaussianWorld) by evaluating on SurroundOcc, a contribution to the broader field of 3D occupancy prediction cannot selectively ignore established community benchmarks.
> > >
> > > The vast majority of influential works in this area (e.g., FB-Occ, SparseOcc, OPUS, GDFusion, COTR) report results on the Occ3D benchmark for nuScenes. Avoiding a comparison on this benchmark makes it impossible to fairly assess the proposed method's performance against the SOTAs and significantly weakens the paper's empirical claims. Furthermore, leading methods in the field (e.g., TPVFormer, OPUS, GDFusion) often provide results on other diverse datasets, such as SemanticKITTI or Waymo, to validate the robustness and wider applicability of their approach.

---

### Note · Authors · 2025-11-24

I have read and agree with the venue's withdrawal policy on behalf of myself and my co-authors.